# Identification of RNA-Binding Proteins as Targetable Putative Oncogenes in Neuroblastoma

**DOI:** 10.3390/ijms21145098

**Published:** 2020-07-19

**Authors:** Jessica L. Bell, Sven Hagemann, Jessica K. Holien, Tao Liu, Zsuzsanna Nagy, Johannes H. Schulte, Danny Misiak, Stefan Hüttelmaier

**Affiliations:** 1Institute of Molecular Medicine, Sect. Molecular Cell Biology, Martin Luther University Halle-Wittenberg, Charles Tanford Protein Center, 06120 Halle Saale, Germany; sven.hagemann@medizin.uni-halle.de (S.H.); danny.misiak@medizin.uni-halle.de (D.M.); 2Children’s Cancer Institute Australia, Randwick, NSW 2031, Australia; tliu@ccia.org.au (T.L.); ZNagy@ccia.org.au (Z.N.); 3St. Vincent’s Institute of Medical Research, Fitzroy, Victoria 3065, Australia; jholien@svi.edu.au; 4Biosciences and Food Technology, School of Science, College of Science, Engineering and Health, RMIT University, Melbourne, Victoria 3053, Australia; 5School of Women’s & Children’s Health, UNSW Sydney, Randwick, NSW 2031, Australia; 6Department of Pediatric Oncology/Hematology, Charité-Universitätsmedizin Berlin, 10117 Berlin, Germany; johannes.schulte@charite.de; 7German Consortium for Translational Cancer Research (DKTK), Partner Site Charité Berlin, 10117 Berlin, Germany

**Keywords:** DICER, inhibitor, LIN28B, *MYCN* amplification, N-Myc, neuroblastoma, RBM, ribosome, RNA-binding protein, TERT

## Abstract

Neuroblastoma is a common childhood cancer with almost a third of those affected still dying, thus new therapeutic strategies need to be explored. Current experimental therapies focus mostly on inhibiting oncogenic transcription factor signalling. Although LIN28B, DICER and other RNA-binding proteins (RBPs) have reported roles in neuroblastoma development and patient outcome, the role of RBPs in neuroblastoma is relatively unstudied. In order to elucidate novel RBPs involved in *MYCN*-amplified and other high-risk neuroblastoma subtypes, we performed differential mRNA expression analysis of RBPs in a large primary tumour cohort (*n* = 498). Additionally, we found via Kaplan–Meier scanning analysis that 685 of the 1483 tested RBPs have prognostic value in neuroblastoma. For the top putative oncogenic candidates, we analysed their expression in neuroblastoma cell lines, as well as summarised their characteristics and existence of chemical inhibitors. Moreover, to help explain their association with neuroblastoma subtypes, we reviewed candidate RBPs’ potential as biomarkers, and their mechanistic roles in neuronal and cancer contexts. We found several highly significant RBPs including RPL22L1, RNASEH2A, PTRH2, MRPL11 and AFF2, which remain uncharacterised in neuroblastoma. Although not all RBPs appear suitable for drug design, or carry prognostic significance, we show that several RBPs have strong rationale for inhibition and mechanistic studies, representing an alternative, but nonetheless promising therapeutic strategy in neuroblastoma treatment.

## 1. Introduction

Neuroblastoma is a common childhood cancer of the sympathetic nervous system [1]. Current treatment strategies for high-risk neuroblastoma are radiation, cytotoxic drugs including topoisomerase inhibitors, surgery, immunotherapy, and retinoic acid [1]. Although more than two-thirds of children do survive, these statistics do not acknowledge the high incidence of chronic health conditions and adult cancer suffered by survivors of high-risk neuroblastoma [2]. Thus, new treatments are desperately needed to both save and improve the lives of these children. Neuroblastoma tumours are characterised by recurrent chromosomal abnormalities including 17q gain, 1p loss and *MYCN* amplification [1]. MYCN protein is expressed in most neuroblastoma tumours and significantly influences cell phenotype and patient outcome. Direct targeting of MYCN has proved non-feasible thus far [3], therefore clearly new drug discovery strategies are warranted. Transcriptomic characterisations of neuroblastoma tissues and chromatin immunoprecipitation sequencing in cell lines have identified several key mRNAs, such as MYCN, which work in transcriptional positive feedback loops [4]. Accordingly, pre-clinical studies assessing therapeutic strategies using transcription modulators such as BET, in addition to CDK7 inhibitors are ongoing. Some investigations are already published [5,6]. The involvement of RNA-binding proteins (RBPs) in these signalling networks has been largely ignored. At the post-transcriptional level, RBPs likely support the transcriptional positive feedback loops, possibly influencing the stability of the key RNAs directly. 

Historically, targeting RNAs via small molecules or drug-like compounds has proved exponentially harder than targeting proteins. RNA often has limited tertiary structure, is dynamic, and its surface chemistry is repetitive, making drug discovery approaches difficult [5,7,8]. Regardless, their clinical relevance has led to approximately 30 unique disease-associated mammalian RNAs being successfully targeted with small molecules [9], and only two drugs, tetrazoid and its precursor linezolid, are approved [10,11]. One way to overcome issues associated with the targeting of RNA is to target the proteins which bind and regulate them. Using this approach, scientists can employ the rational drug design methods which have been utilised for almost 50 years for proteins [12,13].

The post-transcriptional level of regulation in MYCN-driven neuroblastoma has been investigated to some extent (especially regarding miRNAs, reviewed in [14]), yet still represents an under-investigated part of neuroblastoma biology, especially concerning RBPs. This is despite technological advancements, such as the next generation sequencing of RBP bound RNAs through cross-linking immunoprecipitation (CLIP) techniques [15], and analyses of methylated RNA modifications (e.g., m^6^A, and its consequential effects on RBP binding, gene expression and cancer cell phenotype [16]). In this paper, we aim to elucidate novel RBPs that warrant experimental work, give recognition to the important role of reported RBPs in neuroblastoma biology, and outline any therapeutic strategies targeting them. It has proven difficult to find publications on RBP roles in neuroblastoma because the RNA-binding roles are often not investigated or even acknowledged in many oncology publications. Therefore, we chose to anchor those publications we could find, within the wider context of RBP expression analysis in neuroblastoma tumours. We used the R2 visualisation platform (https://r2.amc.nl) to analyse differential gene expression and prognostic values of RBPs from a large primary tumour cohort. Significant clinical grouping association and/or prognostic value were elucidated for more than 600 RBPs, most of these completely novel in neuroblastoma. These novel and reported RBPs are indicative of tumour cells coping with high proliferation rates/replication stress, and a downregulation of genes associated with neuronal homeostasis. We gave more attention to putative oncogenic RBPs, as these could offer a more direct therapeutic target in neuroblastoma.

## 2. Results and Discussion

Using a large neuroblastoma cohort expression dataset (Gene Expression Omnibus (GEO) accession: GSE49710, R2 platform: SEQC-498 custom cohort), we determined differentially expressed RBPs between clinical groupings of interest (Figure 1 and Appendix A). The RBP gene list was defined by Gerstberger and colleagues [17]. Stages 1–3 represent mostly localised, low-risk disease [18]. Stage 4 is the most high-risk stage, highly metastatic with the highest prevalence of *MYCN* amplification. Stage 4S exists in young infants and is a highly metastatic disease with a high incidence of spontaneous regression and good prognosis. A second, smaller cohort (Versteeg, GEO accession: GSE16476 [6]) was analysed using similar methods to above and yielded comparable results (Appendix A). 

### 2.1. There Are Many Novel and Reported RBPs Associated with MYCN-Amplified Neuroblastoma

Firstly, we investigated RBPs relevant to *MYCN*-amplified (MNA) neuroblastoma. Considered mRNAs had to be expressed in at least 100 of the 493 tumours with known *MYCN* amplification status (five tumours were excluded because there were no patient data on *MYCN* amplification status). These tumours were visualised by volcano plot with Kruskal–Wallis rank-based test (Kruskal) and false discovery rate (FDR), with the corrected *p*-values versus the expression fold change between non- and MNA tumours displayed (Figure 1A). 1103 RBPs were significantly different (FDR corrected *p* < 0.05) in expression, either positively or negatively associated (Appendix A).

#### 2.1.1. Ribosomal Proteins: RPL22L1, RPL39, RPL18A, RPL36, RPL35, RPS3

Our analysis shows that several ribosomal proteins are significantly upregulated in MNA neuroblastoma tumours, indicating hyperactive ribosome biogenesis. This is consistent with previous reports in other neuroblastoma contexts [19,20]. Hyperactive ribosome biogenesis is critical in oncogenesis and results in dysregulation of the DNA damage response, altered transcriptional programmes and evasion of T cell-mediated cytotoxicity and other changes [21]. RPL22L1 is the clear outlier of all analysed RBPs, with close to a 2 log-fold difference and being associated with MNA tumours (Figure 1A). RPL22L1 still ranks first by *p*-value, after MYCN mRNA, even with all mRNAs included (*p* = 4.35 × 10^−74^, plot not shown). RPL22L1 has not been investigated in neuroblastoma, however it is over-expressed in metastasising and invasive ovarian tumour tissues, corresponding with the increased DNA copy number [22]. Although originally located on the 3q26.2 locus, the amplified *RPL22L1* gene is co-located with amplified *MYCN* DNA in double minute chromosomes in the UACC-1598 ovarian cancer cell line. *In vitro* enforced expression of RPL22L1 increased tumorigenicity and mesenchymal markers. In our analysis, RPL35 was also associated with MNA tumours and reportedly plays an important role in neuroblastoma [23]. Specifically, RPL35 protein bound lncNB1 (a long non-coding RNA) and E2F1 mRNA and facilitated E2F1 mRNA translation into protein, leading to increased *DEPDC1B* gene transcription. The GTPase-activating protein DEPDC1B also promotes ERK protein phosphorylation, leading to MYCN protein phosphorylation and stabilisation. RPL35 thereby up-regulates E2F1 and MYCN oncoprotein expression and induces neuroblastoma cell proliferation. Importantly, high levels of RPL35 expression in neuroblastoma correlated with poor patient prognosis. In our analyses, RPA18A, RPL39, RPL36, RPS3 were also significantly upregulated in MNA tumours, however, we could not find any literature linking these to neuroblastoma. RPL22L1 and other ribosomal proteins were also associated with MNA in a second neuroblastoma cohort when analysed by similar means (Appendix A).

#### 2.1.2. Ribosome Biogenesis: NPM1 and FBL

NPM1 (nucleophosmin/B23) was the next most significant RBP after RPL22L1 by p-value. Its presence as an MNA-associated RBP is unsurprising as it is reportedly an exemplary MYC/MYCN transcriptional target gene [24]. Depletion of MYCN from the *NPM1* promoter region resulted in a decrease in elongating RNA Pol II and subsequent loss of NPM1 mRNA in neuroblastoma cells. In other contexts, the *NPM1* locus can translocate with *ALK* and the protein preferentially binds G-quadruplex forming nucleic acids [25]. Moreover, it is involved in the biogenesis of ribosomes, and controls apoptotic response to stress and oncogenic stimuli. It is unknown if NPM1 shares these functions in neuroblastoma. Remarkably, in silico screening and experimental studies found that the small molecule, NSC348884, disrupts NPM1 oligomerization, causing apoptosis and DNA damage on its own and synergises with doxorubicin treatment in cancer cells [26]. FBL (fibrillin), also involved in ribosome biogenesis, was significantly associated with MNA status (Figure 1A). FBL is a component of the nucleolar small nuclear ribonucleoprotein (snRNP) particle thought to participate in the first step in processing pre-ribosomal RNA [27]. In neuroblastoma, FBL has been elucidated in an MYC(N)-driven neuroblastoma tumour gene signature [28], but it has not been mechanistically studied. FBL is a therapeutic target and biomarker for using the rRNA synthesis inhibitor, CX-5461, as part of a strategy to inhibit hyperactive ribosome biogenesis [27]. In neuroblastoma, CX-5461 suppressed *MYCN* expression, induced DNA damage, and reduced growth of established MNA neuroblastoma xenografts [20]. In the second neuroblastoma tumour cohort, NPM1 and FBL were highly associated with MNA tumours (Appendix A).

#### 2.1.3. RBPs Involved in Replication Stress and Telomere Maintenance: RNASEH2A, EXO1, TERT, DDX1, DKC1, GAR1, BRCA1 and PARP1

MYCN is an oncogene that is reported to cause severe oncogenic stimuli or replication stress [29,30,31], even apoptosis if overexpressed in other cellular contexts [32]. As expected, factors that cope with replicative stress are upregulated in MNA tumours, such as EXO1, RNASEH2A, DDX1, GAR1, DKC1 (Figure 1A). These mRNAs are also increased in stage 4 tumours (Figure 1B). BRCA1 and PARP1 are also associated with responses to replication stress, and display a similar trend to the above-mentioned RBPs, albeit with less significance and magnitude. The activity of RNase H genes prevents the accumulation and persistence R-loops (RNA:DNA hybrid structures) [33]. If not resolved, R-loop structures hinder replication, modulate transcription and influence genomic instability. EXO1 has both 5’ to 3’ exonuclease activity as well as an RNase H activity, and resolves stalled DNA replication, thus inhibiting replication stress and potential replication fork collapse [34]. Although often highly expressed in cancer cells, its role in neuroblastoma remains unstudied. Diallyl disulphide (DADS), a compound found in garlic, has shown therapeutic potential in reducing EXO1 levels and in turn inhibiting DNA resection and repair, though its exact mechanism remains unknown [35]. RNASEH2A is also novel for neuroblastoma, however putative HIV RNASEH2 drugs (RHI001 and RHI002) have been shown to inhibit human RNASEH2 and decrease cell numbers of the neuroblastoma cell line, SK-N-SH [36]. Interestingly, over half of MNA neuroblastoma tumours have co-amplified *DDX1* DNA (on the 2p locus) [37]. Although DDX1 (an RNA helicase) has reported functions in replication stress (DNA damage), RNA fate, and pre-microRNA processing [38], no mechanistic studies are published in neuroblastoma. Perhaps of current interest, human DDX1 is a critical factor in the genomic RNA replication of corona viruses and a drug target for that disease [39].

Three recurrent genetic aberrations in neuroblastoma, *MYCN* amplification, *TERT* re-arrangements, and *ATRX* mutations, promote telomere maintenance mechanisms and are mutually exclusive mutations [40]. TERT is an RBP and is significantly increased in both the MNA and stage 4 tumour groupings (Figure 1). Notably, children with stage 4S neuroblastoma, which is known to spontaneously regress half the time, show no TERT protein expression [41]. TERT activation by genomic rearrangement is also reported in neuroblastoma [42,43,44]. TERT silencing, or treatment with 6-thio-2′-deoxyguanosine in melanoma, leads to rapid cell death and both telomeric and non-telomeric DNA damage, and offers an untested therapeutic strategy in neuroblastoma [45]. Aberrant tumour expression of other telomerase holoenzyme components including DKC1 (dyskerin), NHP2, and GAR1 in tumours was reportedly associated with poor prognosis in neuroblastoma patients [39]. The knockdown of DKC1 expression reduces TERC expression and telomerase activity. Furthermore, in neuroblastoma, DKC1 has been reported as an MYC/MYCN transcriptional target, with its downregulation halting cell replication, suppressing tumour formation and inducing ribosomal stress [46]. Notably, DKC1 inhibitors have been elucidated via computational modelling methods and show anticancer activity [47,48]. 

BRCA1 is generally considered a tumour suppressor involved in DNA repair. However, a recent report outlines a clear oncogenic-like role for BRCA1 in enhancing transcriptional activation by MYCN [29]. MYCN was found to recruit BRCA1 to the promotor-proximal DNA where R-loops arise, thereby promoting mRNA decapping and R-loop resolution. Depletion of BRCA1 slowed cell cycle progression and increased DNA damage. Its high expression in neuroblastoma tumours was associated with decreased patient survival. BRCA1 protein also inhibited MYCN proteasome turnover. Additionally, BRCA1 regulates miRNA biogenesis via binding to the DROSHA microprocessor complex and RNA helicase DHX9 [36]. This function was described in other cancer types and may also have relevance in neuroblastoma. 

PARP1 inhibitors are a therapeutic strategy for BRCA1/PARP pathway de-regulation. Notably, neuroblastomas with deletion of 11q might be a therapeutic target for PARP inhibition [49]. PARP1 (an RBP) is also reported as highly expressed in neuroblastoma and is associated with reduced patient survival [30]. PARP1 is induced by *MYCN* expression and PARP inhibitors enhance the replication stress caused by MYCN in neuroblastoma cell lines. 

In the second cohort, many of the above factors were also strongly associated with MNA tumours, especially EXO1 and DDX1 (Appendix A).

#### 2.1.4. Other Reported MYCN-Associated RBPs: NONO, IGF2BP1, IGF2BP3, and EZH2

There are several other reported RBPs associated with MNA in neuroblastoma which may influence *MYCN* expression. NONO (p54nrb) binds to NCYM (the lncRNA which is transcribed from the strand opposite of the *MYCN* gene) [50], and also binds to lncUSMycN, (the lncRNA which is transcribed upstream of the *MYCN* and *NCYM* genes) [51]. NCYM and the lncUSMycN lncRNA enhance NONO-mediated MYCN mRNA stability and up-regulation [50,51]. In addition, NONO interacts with the internal ribosome entry sites (IRES) of MYCN and MYC mRNAs, and thereby increases MYC(N) mRNA translation into protein [52,53]. In human neuroblastoma tissues, high levels of NONO mRNA correlate with high levels of MYCN mRNA expression, and this correlation predicts poor patient prognosis [51]. Last year, NONO was found to suppress RNA-telomere instability in mouse embryonic stem cells, but this role remains unexplored in neuroblastoma [54]. Although no inhibitors are available for NONO, the YM155 compound (which inhibits BIRC5/survivin transcription) has been described to disrupt the NONO interaction with ILF3 and decrease NONO abundance [55].

IGF2BP family members (IMPs) are highly expressed in cancers and are required for the migration of neural crest cells during development, modulation of neurite outgrowth, and regulation of stem-cell properties centrally within the LIN28/let-7 signalling network [56,57]. The binding of IGF2BPs to RNA relies on the m^6^A RNA epi-transcriptomic modification [58]. All family members are reported to be expressed in up to 100% of neuroblastoma tumours, with high IGF2BP1 expression associated with poor overall prognosis and *MYCN* amplification [59], as well as MNA cell lines [23]. In our analyses, IGF2BP1 was highly expressed in MNA tumours, though the significance was lower than the top elucidated candidates discussed here. IGF2BP3 displayed no clear associations, and IGF2BP2 was highly expressed in stage 4S (Figure 1A,B). In neuroblastoma, IGF2BP1 is reportedly gained at the DNA level, likely as part of a 17q chromosomal gain [59]. siRNA knockdown of IGF2BP1 results in reduced *MYCN* expression and reduced cell viability. High expression of IGF2BP3 in neuroblastoma is reported in two studies [60,61], the latter showing depletion of IGF2BP3 results in reduced cell number. Interestingly, in drosophila embryos, *dIMP* (Orthologue of IGF2BP1/2/3) expression was found to be required for the transcription factor chinmo’s induction of neuroblast tumour growth [62]. Therapeutic agents do exist against IGF2BPs. The growth of IGF2BP3-driven thyroid tumour models was blocked by IGF1R (OSI-906) inhibition [63], and in adult cancer cells, the small molecule BTYNB inhibited IGF2BP1–RNA interactions [64]. Of note, this protein family has several reports regarding autoantibodies against them produced in adult cancer patients (e.g., [65,66]).

Reported as having activity in MNA neuroblastoma cells, the polycomb repressive complex 2 (PRC2) component protein, EZH2, has also been shown to directly interact with lncRNAs, such as NBAT1. NBAT1 enhances EZH2-mediated transcriptional silencing of key tumour suppressor genes, leading to neuroblastoma cell differentiation block and cell proliferation [67]. While EZH2 is a well-known tumorigenic factor and therapeutic target for neuroblastoma, EZH2 is likely to exert tumorigenic functions through both RNA-binding-dependent and -independent pathways. EZH2 is described as an MYCN transcriptional target in erythroleukemia, and the aurora kinase inhibitor MLN8237 was used in this context to repress the MYCN–EZH2 axis [68]. Currently, the EZH2 inhibitor tazemetostat is under clinical evaluation for use in recurrent neuroblastoma (Phase II, Clinical Trials Identifier: NCT03213665).

NONO and IGF2BP1 were also significantly associated with MNA in the second tumour cohort (Appendix A).

### 2.2. RBPs Associated with Lower-Risk Groupings and Involved in Neuronal Homeostasis and Splicing: ELAVLs, CPEBs, CELFs, TDRDs, ZCCHC17, ATXN1/2 and RBFOX1

Although likely not drug targets directly, it is worth noting the RBPs associated with less aggressive tumours. ELAVL2-4 and members of the CBEP, CELF, and TDRD protein families are significantly lower in abundance in MNA tumours, perhaps reflecting a lower proportion of differentiated neuronal cells in those tumours (see Figure 1A, Appendix A). ELAVL2-4 (HuB, C, D) proteins are neuronal-specific [69] and are downregulated in MNA tumours. ELAVL4 (HuD) had the most significant differential expression of its family (*p* = 1.6 × 10^−26^). ELAVL2–4 are necessary and sufficient to induce neuronal differentiation [69], and are potential diagnostic markers of neuroblastoma (distinguishing neuroblastoma from similar tumours) [70]. They promote the stabilisation and/or translation of transcripts containing AU-rich elements (ARE), and Lazarova and colleagues found that ELAVL4 binds to the 3´UTR of the MYCN mRNA, stabilising it [71]. This was subsequently explained by an overlapping ELAVL4 and miR-17 binding site on the MYCN 3’UTR [72]. Surprisingly, the downregulation of ELAVL4 in non-amplified cell lines decreased *MYCN* expression, but increased *MYCN* DNA copy number. This was the opposite effect of that observed in the MNA cell lines [73]. It is therefore suggested that a genetic imbalance of ELAVL4, miR-17 and MYCN might select for *MYCN* amplification and explain the co-occurrence of 1p loss (including the ELAVL4 locus), which is common in MNA tumours. ELAVL1 (HuR), in contrast to its family members, is upregulated in MNA tumours (*p* = 5.7 × 10^−7^ – Appendix A), and also not well studied in neuroblastoma. Notably, one report describes it as protecting against oxidative stress in neuroblastoma cells [74]. Another shows high expression of the circAGO2 RNA in neuroblastoma tumours and cells [75]. It has been demonstrated that this circular RNA binds and promotes ELAVL1-directed protection of transcripts from miRNA downregulation, thus, this is possibly one mode by which ELAVL1 acts in neuroblastoma. Although the rationale for inhibitors against ELAVL proteins in neuroblastoma is unclear, it has been shown in glioblastoma cancer models that the ELAVL1 small molecule inhibitor MS-444 resulted in induction of apoptosis and decreased invasive potential [76]. RBFOX1 had the largest fold difference in non-MNA and stage 4S groupings. Underwood et al. showed that RBFOX1 (Fox-1) has high expression in brain tissue and the neuroblastoma cell line LAN-5, with subsequent in vitro studies showing that RBFOX1 enhances neuron-specific splicing patterns in mouse N2A mRNAs [77].

Appendix A shows more clearly CBEP, CELF, and TDRD family members as being significantly associated with non-MNA tumour status, with CBEP4 along with ZCCHC17 being the most significantly changed mRNAs in non-MNA neuroblastoma tumours. All these above gene families have reported functions in neuronal diseases (such as Alzheimer’s, autism, and epilepsy) and may reflect splicing changes and more neuronally differentiated or functioning cell types in those tumours. ZCCHC17 is a regulator of synaptic gene expression in Alzheimer’s disease, but was also reported to cause aberrant mRNA splicing and inhibits cytoplasmic localisation of RNAs [78]. CPEB family members regulate the balance between senescence and proliferation via influencing translation of specific mRNAs and poly(A)-tail length [79]. CPEB4 and RBFOX1 are both highly expressed in autism, both being important splicing factors in neuronal cells and epilepsy since they promote each other’s expression. It is unsurprising that they are associated together in our analysis. CPEB1 and CPEB2 were reported to bind and regulate the HIF1A 3’-UTR following insulin stimulation in neuroblastoma cells [80]. It remains unknown if MYCN mRNA contains the cytoplasmic polyadenylation element (CPE) that CPEBs bind. All CELF proteins influence pre-mRNA splicing and translation, many proteins having neuron-specific functions [81]. In cancer cells, CELF6 inhibits cell cycle progression and cell proliferation in a TP53 and/or CDKN1A(p21)-dependent manner, and can bind the 3′UTR of CDKN1A, increasing its CDKN1A transcripts stability. Lastly, many members of the Tudor domain-containing protein (TDRD) family were lower expressed in MNA tumours, especially TDRD7, TDRD12, TDRD9 (Figure 1A and Appendix A). TDRD10 was also higher expressed in the 4S tumour grouping compared to stage 4 tumours (Figure 1B). TDRD proteins mainly recognize the N-terminal arginine-rich motifs of PIWI proteins and have roles in adult cancer, eye development and spermatogenesis, and require characterisation in neuroblastoma [82]. Of note, other spicing factors SNRPF and SNRPD1, in contrast to the proteins discussed above, were significantly associated with MNA in tumours (with SNRPF also associated with stage 4). This suggests that splicing in MNA neuroblastoma is likely altered by both increases and decreases in splicing factor abundance.

ATXN1 and ATXN2 mRNA was associated with non-amplified neuroblastoma and ATXN1 was associated with 4S Neuroblastoma and better patient outcome (Figure 1 and Appendix A). Both genes are associated with neurodegenerative diseases, and in neuroblastoma, ectopic expression of ATXN2 sensitizes *MYCN* expressing cells to apoptosis [83]. Tau protein (MAPT) was not significant in any of our analyses, although one report found that Tau expression in neuroblastoma correlated with increased overall survival in neuroblastoma [84]. Tau is also a well-reported RBP in neurodegenerative diseases.

Many of the RBPs mentioned above as being associated with the lower-risk clinical groupings in the SEQC cohort were also found to be differentially expressed in the second tumour cohort (Appendix A). 

### 2.3. There Are Several Novel and Reported Genes that are Expressed with Significant Differences for Stages 4 and 4S Comparisons: PTRH2, TRMT112, KHDRBS2, MKRN3, RBM11, RBMX2, NXF2 (NXF1)

Secondly, we investigated RBP expression in stage 4 neuroblastoma in comparison to stage 4S. Both stages being highly metastatic, stage 4 is the main focus of treatment development, as 4S has a high-rate of spontaneous regression and carries a much lower risk of death. This comparison might indicate how to force regression in stage 4 disease. Although there were shared differentially expressed RBPs with the previous analysis, we did see unique trends. A total of 674 RBPs had a significant difference in expression (FDR corrected *p* < 0.05, Appendix A, Figure 1B). The top two genes based on p-value with higher expression in stage 4 were PTRH2 (*p* = 1.6 × 10^−10^) and TRMT112 (*p* = 5.2 × 10^−10^). PTRH2 (located on Chr. 17q) is associated with an infantile-onset genetic multisystem neurologic, endocrine, and pancreatic disease (IMNEPD), with noted peripheral neuropathy [85]. It prevents the accumulation of dissociated peptidyl-tRNA, which could inhibit protein synthesis, and it influences cell survival and is unreported in the context of neuroblastoma. TRMT112, also novel in neuroblastoma, acts as an activating platform of four *S*-adenosyl-l-methionine (SAM)-dependent methyltransferases modifying rRNA (Bud23), tRNAs (Trm9 and Trm11) or the eRF1 class I translation termination factor (Mtq2), and it has roles in ribosome biogenesis and cell proliferation [86].

The stage 4-associated RBPs that had the largest fold change between the compared stages were RBM11, NXF2 (NXF1), KHDRBS2, RBM20, MKRN3 (Figure 1B and Appendix A). Several RBM family members were represented as either associated with stage 4 or MNA neuroblastoma to a lesser extent, including the RBM11, RBM20, RBMX, RBMX2 (very high significance). These proteins belong to a huge and diverse family of glycine-rich proteins containing one RNA recognition motif (RRM) protein domain and an adjacent arginine and glycine-rich domain. RBM11 is a known tissue specific splicing factor (in the brain, cerebellum, and testis) [87]. Furthermore, it switches splicing of MDM4 and CCND1 to more oncogenic isoforms [88]. The highest differentially expressed RBM in MNA versus single tumours was RBMX. RBMX has roles in neurite growth promotion, splicing [89], as well DNA damage response and DNA replication by interacting with the lncRNA NORAD, topoisomerase I (TOP1), ALYREF and the PRPF19–CDC5L complex [90]. RBMX2 is required during early development with mutants resulting in increased cell death and decreased differentiated neurons, and it was elucidated as a putative splicing factor in neuroblastoma [91]. There are indications that RBM20 also functions in aberrant splicing in cardiomyopathy [92]. In contrast to the RBM proteins discussed above, RBM47 is significantly lower expressed in MNA and in stage 4 (Figure 1). RBM47 is a known tumour suppressor in breast and colon cancer and was recently reported as a regulator of DNA-damage-induced p53 and p21WAF/CIP1 expression [93]. The above RBM genes require characterisation in neuroblastoma. Notably RBM20 was one of the top differentially expressed genes in the second cohort according to fold-change (Appendix A). RBMX2 was also significantly associated with stage 4. 

Although not a top differentially expressed RBP in our analyses, RBM3 has reported roles (many in neuroblastoma cell lines) in protein synthesis and miRNA levels in response to cellular stress conditions such as cold-shock and hypoxia [87]. Specifically, in SH-SY5Y cells, RBM3 possesses neuroprotective effects against various neurotoxins, such as rotenone [94], 1-methyl-4-phenylpyridinium [95], nitric oxide [96], and UV irradiation-induced apoptosis [97]. Notably, RBM3 promotes neurogenesis after hypoxic-ischemic brain injury in a niche-dependent manner via IGF2BP2-IGF2 signalling [98]. Although RBM3 is a proto-oncogene (in that it facilitates cell division and attenuates apoptosis), its expression is surprisingly associated with favourable oesophageal cancer phenotype in tissue microarrays [99].

Of the remaining novel candidates, NXF1 (of which NXF2 is a splice variant) has roles in transcriptional dynamics, 3’ end processing, and nuclear export of long 3’ UTR transcripts, implicating NXF1 as a nexus of gene regulation [99]. MKRN3 is a novel imprinted gene, encoding a RING zinc-finger protein. Deleterious mutations of *MKRN3* and lower expression levels in the brain and serum are implicated in early puberty [100]. In the hypothalamus, *MKRN3* expression inhibits puberty and is negatively regulated by miR-30b [101]. In addition, mass spectrometry analysis detected that MKRN3 interacts with IGF2BP2 and LIN28B [102]. KHDRBS2 (SLM1) was also an outlier in our analyses, being associated with stage 4 neuroblastoma. It regulates cell-type specific alternative splicing of NRXN1 and acts synergistically with KHDRBS1 (Sam68) in exon skipping [103]. KHDRBS2 remains novel in cancer, however, high protein expression of its homologous paralogue KHDRBS1 was associated with decreased survival probability in neuroblastoma [104].

### 2.4. RBPs Associated with High-Risk Patient Death: AFF2 and BARD1

In order to elucidate putative oncogenic RBPs that could represent biomarkers or novel drug targets in cases where the conventional neuroblastoma treatments fail, we produced a volcano plot using high-risk patient tumours only and compared those tumours where the patients died versus survived (Figure 2 and Appendix A). A total of 120 RBP mRNAs yielded significant *p*-values (FDR corrected <0.05). AFF2 (FMR2 Family Member 2) is a clear outlier of importance, the rest of the factors having only slight magnitudes of difference. AFF2 was also significantly increased in the stage 4 neuroblastomas versus 4S (Figure 1B). *AFF2* mutations are implicated in X-linked intellectual disability [105]. AFF2 is a nuclear transcriptional activator that encourages RNA elongation and influences biogenesis of nuclear speckles; likely influencing R-loops, splicing efficiency of G-quadruplex structures, and potentially expansion of GGGGCC repeats [105,106]. Additionally, BARD1 was another RBP significantly increased in patients that died. Interestingly, BARD1 polymorphisms convey increased risk of neuroblastoma and correlate with increased expression of an oncogene isoform (BARD1β) which lacks a RING domain, interacts with AURKB and blocks apoptosis [107]. In other cell models, BARD1 interacts with BRCA1 via its RING domain and is involved in DNA damage response. Indirect drugs targeting cells with BARD1 deregulation include PI3K and PARP inhibitors [108,109].

### 2.5. One-Third of RBPs Have Prognostic Value in Neuroblastoma, Including MRPL11, RAN and HNRNPC

To further assess whether any oncogenic-like RBPs are valid drug targets in neuroblastoma, we performed a Kaplan–Meier scan analysis in the neuroblastoma SEQC cohort using the RBP gene list from Gerstberger et al. [17]. From the 1483 RBPs analysed, 685 had prognostic value in patient event-free survival, with either low or high RBP expression being associated with poor prognosis (Appendix A). Figure 3 shows the top 30 mRNAs. Notably, some group sizes are small and not all these RBPs would retain significance upon median or other cut-offs (we have corrected for multiple testing). The most significant gene by p-value was *MRPL11* (a mitochondrial ribosomal protein), which is also more highly expressed in the stage 4 tumours (Figure 1B) but is yet to be studied in neuroblastoma. In head and neck cancer cells, MRPL11 regulates mitochondrial translation and oxidative phosphorylation [110]. Tigecycline, an FDA-approved broad-spectrum antibiotic, selectively targets ovarian cancer cells and could be a therapeutic indirect strategy against de-regulated MRPL11 in neuroblastoma [111]. The aforementioned GAR1, EXO1, SNRPF, DKC1, FBL, RNASEH2A, SNRPD1, RPL39 and NPM1 were also mRNAs with high expression corelating with poor prognosis in the top 30 list. Of the top 30 mRNAs, RAN is perhaps the most reported RBP in cancer. In neuroblastoma, *RAN* expression is promoted by LIN28B and *RAN* expression is higher in 12q24 chromosomal region gain tumours (where it is located) [112]. RAN also promotes AURKA activity, with Schnepp and colleagues suggesting that RAN overexpression could be targeted by aurora kinase inhibitors. The RAN inhibitor, NSC311153, an ellipticine derivative, could also be tested in neuroblastoma tumours with *RAN* expression. *RAN* was significantly higher expressed in MNA and stage 4 groupings in our analyses grouping comparisons (Appendix A).

HNRNPC was ranked 30 in our Kaplan scan analyses. Overexpression of HNRNPC has a critical role in the establishment of alternative cleavage and polyadenylation profiles, which are characteristic of metastatic colon cancer cells [112]. Furthermore, the loss of HNRNPC strongly reduces the abundance of BRCA1, BRCA2, RAD51 and BRIP1 due to aberrant splicing effects [113]. In neuroblastoma cells, HNRNPC was shown to promote APP translation by displacing the FMRP protein [114]. Interestingly, other HNRNP family members reached significance in our various analyses, e.g., A1, C, K, F, U and D (see Appendix A). HNRNPA1, although not a top gene in our analysis, did reach statistical significance in MNA tumours and in prognosis analyses. Notably, MYCN is reported to control global splicing programs via genes including HNRNPA1, with HNRNPA1 knockdown repressing cell growth [115]. In another study, the small molecule VPC-80051 was developed to target the RNA-binding domain (RBD) of HNRNPA1 and inhibit its splicing activity [116]. Furthermore, the antihypertensive drug amiloride downregulates HNRNPC and other family members by modulating genome-wide alternative splicing, including altering BIRC5 (survivin) splicing [117].

### 2.6. Other RBPs Reported in Neuroblastoma Involved in miRNA Biogenesis: LIN28B, DROSHA, DICER and DHX9

Many factors already reported in neuroblastoma and involved in miRNA biogenesis were not top factors significantly associated with clinical groupings in our analyses. However, these proteins are important in early genetic events and initiation of the cancer and may have roles in all neuroblastoma cells. The RBP LIN28B is a driving oncogene of neuroblastoma and a central regulator of neuroblastoma tumour biology. LIN28A, a paralogue of LIN28B, was identified as a member of an essential set of four genes able to reprogram somatic cells into induced pluripotent stem cells [118]. Furthermore, *LIN28A* and *LIN28B* are important developmental genes. In line with a role in neuroblastoma initiation, LIN28B is involved in the regulation of neural crest precursor cell proliferation and differentiation [119]. It was also among the first RBPs shown to regulate miRNAs, via the binding and downregulation of let-7 family members [120]. One of the first studies suggesting a role for LIN28B in neuroblastoma was a genome-wide association study (GWAS) that identified the LIN28B SNP rs17065417, located in the intronic regions of the *LIN28B* gene to be correlated with increased neuroblastoma risk [121]. In addition, focal amplifications of the *LIN28B* locus were detected in a minor, yet significant, fraction of neuroblastoma tumours [122,123,124]. Importantly, high expression of LIN28B correlates with adverse outcome in neuroblastoma [121,124].

Subsequently, additional mechanisms resulting in the deregulation of LIN28B in neuroblastoma were reported, including the downregulation of the LIN28B mRNA targeting miRNAs mir-125a [125] and miR-26a/b-5p [126], which correlated with an upregulation of LIN28B by MYCN [126,127]. Unsurprisingly, downregulation of LIN28B in MNA neuroblastoma cells reduced cell viability [121,124] and, most significantly, overexpression of LIN28B in the neural crest cells of transgenic mice resulted in neuroblastoma formation *in vivo* [124]. This proved that LIN28B is a bona-fide neuroblastoma oncogene. Furthermore, these data are supported by experiments demonstrating *LIN28B* expression results in the transformation of trunk sympathoadrenal precursors via suppressing let-7, and in turn increasing *MYCN* expression [128]. As MYCN also upregulates LIN28B, this indicates that a clinically significant positive feedback loop exits between the two proteins in neuroblastoma. In addition to miRNA binding, LIN28B directly binds mRNA and long non-coding RNAs. In other cell models using CLIP analysis, LIN28 was found to interact with thousands of transcripts via two binding domains, the zinc knuckle protein domain which binds GGAG-like RNA sequences, and the cold shock protein domain which binds (U)GAU RNA sequences and may be associated with let-7 binding [129]. LIN28B also promotes the expression of the *RAN* oncogene by directly interacting with its mRNA (as discussed earlier [130]). Several attempts have been undertaken to therapeutically target LIN28B–Let7 interactions. In one such screen, the compound 1632 blocked the Lin28/let-7 interaction and rescued let-7 processing and function in *LIN28B*-expressing cancer cells [131], though *in vivo* studies were not performed. Shahbazi et al. showed that BRD4 inhibition (via the small molecule JQ1) in combination with an HDAC inhibitor (panobinostat) effectively suppressed *LIN28B* expression in neuroblastoma cells, suppressing both LIN28B’s miRNA and mRNA-binding effects, and showed *in vivo* anticancer activity [132].

The biogenesis of miRNAs relies upon the RNase III endonucleases, DROSHA and DICER (reviewed in [133]). These proteins are reported as having altered expression and potentially tumour-suppressive activity in neuroblastoma, although they were not greatly changed compared to other RBPs in our analyses. Lower mRNA levels of DICER and DROSHA was reported to correlate with a global downregulation of miRNAs and poor patient outcome [134]. Both *DICER* and *DROSHA* expression were significantly lower in stage 4 disease and MNA tumours when compared to other tumours. Notably, *DICER* expression was an independent predictor for overall survival (hazard ratio, 9.6) in tumours with non-amplified *MYCN*. The mechanisms that regulate expression of *DICER* and *DROSHA* in neuroblastoma remains unclear. However, a common loss of heterogeneity on chromosome 14q [135], which possibly includes the *DICER* locus, could reduce *DICER* expression. Additionally, neuroblastoma is associated with pleuropulmonary blastoma, a syndrome often having *DICER1* mutations [136].

DROSHA and DICER have reported roles in neuroblastoma cell proliferation and differentiation phenotypes. Knockdown of DICER or DROSHA in neuroblastoma cell lines reduced miRNA abundance (quantified by miRNA array), as well as enhanced cell proliferation and clonogenicity in soft agarose [134]. Apart from having a fatal impact on brain development, *DICER* knockout mice also have altered neural crest cell differentiation (the originating cells of neuroblastoma) [137], and in accordance, *in vitro* studies show that DICER is strongly up-regulated in differentiating neuroblastoma cells [138,139], and that its knockdown induces senescence in differentiating SH-SY5Y cells [140]. Interestingly, some neuronally differentiated neuroblastoma cells, including the SK-BE(2)-C cell line, express an alternatively spliced and truncated DICER (t-Dicer) protein, lacking the catalytic residue in its RNase RIIIb domain [141]. T-Dicer was detectable in 29 of 55 primary neuroblastic tumours, but not in the normal and other cancer tissues tested. Additionally, t-Dicer functions less efficiently than full length DICER, but not due to dominant negative effects [142]. Further pre-clinical studies are required to ascertain if DROSHA, DICER, and their variants are therapeutic targets or biomarkers in neuroblastoma. Topoisomerase inhibitors are predicted to inhibit DROSHA and DICER pri-miRNA interactions [143], and additional computer simulations suggest that three metabolites interfere with DICER pri-miRNA-binding [144], but *in cellulo* validation is required. 

DHX9 (RNA Helicase A) binds BRCA1 and influences miRNA biogenesis [145], and it also binds the IGF2BP1 protein influencing MYC RNA stability [146], as well as a reported function in neuronal and neuroblastoma cells [147]. In the latter, induction of KIF1B increases nuclear *DHX9* expression during NGF withdrawal, promoting apoptosis; however when KIF1B is reduced as part of 1p loss (where it is located), DHX9 nuclear localisation is impaired, and cells are protected from apoptosis. DHX9 was also elucidated as a putative biomarker in lung cancer by mass spectrometry of tumours and elevation in patient serum [148]. In the same study, the toxicity of the quinolone drug enoxacin was dependant on *DHX9* expression. DHX9 showed no strong association with our tested clinical groupings but was ranked 565th by prognostic value (high expression associated with worse outcome).

### 2.7. Elucidated and Reported Oncogenic-Like RBPs Are Highly Expressed in Neuroblastoma Cell Lines

To appreciate the relative levels of mRNA expression of top putative RBP oncogenic targets, we produced a heat map of their expression in neuroblastoma cell lines and the tumour subset (Figure 4). We only choose one RBP family member and the RBPs with the most clinical/reported significance in our evaluations to create a list of putative RBP target oncogenes in neuroblastoma. For these candidate oncogenes, we displayed their expression in neuroblastoma cell lines (ordered by MYCN mRNA abundance) with data from the Broad Institute Cancer Cell Line Encyclopedia (CCLE) [149], the SEQC tumour cohort groupings, as well as their cell selectivity upon knockout using the Broad Institute Cancer Dependency Map, CRISPR Avana dataset 17Q4 [150]. We also chose one novel (CEBP4) and one reported tumour-suppressor-like (ATXN2) RBP to provide some contrast to the oncogenic-like factors in the heatmap comparisons. We observed that RPL22L1, FBL, DDX1, NPM1, RPL35, NONO, TRMT112, RAN and HNRNPC are highly abundant at the RNA level in most neuroblastoma cell lines and also in tumours. These genes were also selective upon genomic deletion in all cancer lines (as indicated by low dependency scores), not just in neuroblastoma cell lines, but also in other cell lines to a lesser extent. This supports the rationale for testing/developing inhibitors for these RBPs for use in neuroblastoma. *EXO1*, *IGF2BP1*, *BRCA1*, *LIN28B* and *DICER1* expression in neuroblastoma CCLE cell lines shows a trend towards higher expression when *MYCN* is also highly expressed. LIN28B has a lower dependency score (showing more cell selectivity upon knockout in a population of cells) in neuroblastoma cells compared to other cell lines. As most of the cells analysed are MNA, this is at extreme odds with the publication of Powers et al., which states LIN28B to be dispensable in MNA cell lines, using similar methods [151]. *EZH2* expression is also correlated with *MYCN* expression, however its dependency score suggests it may not be a good candidate for neuroblastoma inhibitor testing or other cancers in general. Notably, *in vivo* effects may differ to the effects in cell lines and the genomic depletion strategies employed in the dependency score dataset. As EZH2 inhibitors exist, and previous reports support its function in neuroblastoma as an oncogene, it would still be worth assessing their efficacy in neuroblastoma cell lines. TERT is known to be activated in highly aggressive neuroblastoma, however, compared to other RBPs, it has low levels of RNA expression. KHDRBS2, MKRN3 and AFF2 were generally lower expressed at the mRNA level compared to the other proteins displayed, and the first two genes are not selective against neuroblastoma cell line proliferation under normal growth conditions upon knockout (regarding dependency scores). *CPEB4* appears to be a tumour suppressor gene of consequence in neuroblastoma cell lines and tumours, which is confirmed by a higher dependency score in neuroblastoma cell lines compared to the other cell lines combined, i.e., neuroblastoma cells grow faster/have greater survival after CRISPR disruption of the CPEB4 DNA. *In vivo* studies and mechanistic studies are warranted to confirm CPEB4 as a tumour suppressor in neuroblastoma, as well as studies elucidating the cause of CPEB4 downregulation and if it is pharmacologically reversable. 

### 2.8. Summary of Top Oncogenic-Like RBP Regarding Available Structures and Inhibitors 

In 2018, the first candidate drug targeting an RBP, H3B-8800, reached clinical trials for the treatment of acute myelogenous leukaemia and chronic myelomonocytic leukaemia [152]. This drug modulates the splicing factor SF3B1 protein. Computational methods, specifically pharmacophores, were utilised to help design the initial drug hits and the structural method of cryogenic electron microscopy explained the drug’s mechanism of action. This compound demonstrated the importance of using structure-based methods to develop inhibitors of RBPs. Importantly, the aforementioned ELAVL1 (MS-444 [76]) and HNRNPA1 (VPC-80051 [116]) inhibitors, and some others discussed in this paper, employed computer-aided discovery of small molecules. Although historically compound screens paired with binding assays have been favoured to develop protein–nucleic acid inhibitors (as for LIN28 [131]), the above recent successes demonstrate that when structural information is known, computer-based drug discovery is a viable method to discover and develop drugs against RBPs. Notably, due to the dynamic and flexible nature of the RNA–protein complex, structure-based drug discovery methods utilising x-ray crystallography cannot be used in isolation and require solution-based binding analyses within the testing pipeline, such as molecular dynamics and machine learning methods to improve the drug design [153].

Excited by the potential of the putative novel and reported oncogenic RBPs in Figure 4, we wanted to understand how potentially “druggable” our top candidates were. Thus, characteristics that are important in RBP drug design are summarized in Figure 5. Specifically, known RBP domains and roles (UniProt website [154]), transcriptome-wide binding informatic data (CLIP studies POSTAR2 tool [108]/NCBI GEO), if the crystal structure is solved (UniProt and CanSAR website [155]), and whether that structure has predicted potential for designing drugs according to the CanSAR website are presented [155]. Please note that information from other family members/orthologues could be used for structure modelling, which we have ignored in our figure. Emerging oligonucleotide-based interference possibilities have also been excluded due to current lack of efficacy in delivering oligonucleotides to neuroblastoma tissues. Many candidate oncogenic RBPs have characteristics suitable to start computer-aided drug discovery (Figure 5), which is very encouraging toward developing therapies for these novel proteins. As mentioned above, several proteins either already have inhibitors, or there are indirect strategies reported, which we have listed in Figure 5. Of course, it is even better if RNA-binding transcripts are well-characterised with cellular phenotypes, however, this is just a simplistic overview. One could also start with a key RNA and use pull down strategies and CLIP searchers to characterise binding to a specific protein or group of proteins and use this toward combination drug design. Not all candidate oncogenic RBPs have structures which could support current drug design programs, and these may have to be targeted indirectly or by an alternative strategy.

## 3. Conclusions

RBPs that are differentially expressed and reported in neuroblastoma, influence almost every step in the development and progression of cancer. In neuroblastoma, we found RBPs which influence death-resistance, proliferation, cell cycle control, replication stress (DNA damage response), oxidative stress, chemosensitivity, splicing and translation in neuroblastoma and related cell types. Ribosome biogenesis and telomere maintenance appear particularly important from an mRNA level perspective. Moreover, RBPs that bound G-rich motifs, or corrected problems associated with G-rich structures (such as R-loops), were highly represented in our candidate oncogenic RBP list. In the more aggressive neuroblastoma clinical groupings, RBPs with reported roles in neuronal function had less abundance (e.g., CPEB4), perhaps indicating a loss of the peripheral neuronal cell identity and function. The contexts of metastasis, autophagy, angiogenesis and immune surveillance, whilst likely areas where RBPs probably have roles in neuroblastoma, remain relatively unstudied and represent a research area with huge potential. We could not find any high-throughput CLIP studies performed in neuroblastoma cells, and this is essential to define the repertoire of binding transcripts in the neuroblastoma context, especially MYCN-driven cells, and it is helpful for drug design and screening. In many reported RBP articles, the RNA-binding mechanism at the post-transcriptional level was often not well investigated. This is also important, as it could allow more efficient drug screening and design if an RNA-binding target with a well-defined functional significance can be used. Both cohorts, the larger SEQC and the second cohort (Versteeg), displayed similar differentially expressed RBPs. We noticed protein paralogues being upregulated together or displaying disparate trends. For example, in the large SEQC cohort, RBM11, RBM20, RBMX2 were all upregulated in stage 4 neuroblastoma, compared to 4S with RBM47 being downregulated. Furthermore, in the MNA tumours, RBFOX3 was upregulated, whereas RBFOX1 was downregulated, perhaps indicating changes in splicing. IGF2BP1 was increased in MNA tumours, whereas in 4S disease (the possibly tumour suppressive) IGF2BP2 was the most up-regulated family member. These related RBPs may compete for binding of the same transcripts yet confer differential effects on the target RNAs’ life cycle. For example, IGF2BP1 and IGF2BP2 have a shared consensus GG(m^6^A)C binding sequence [58], though in adult cancer cells, the binding of IGF2BP1 and not IGF2BP2 results in the protection of some oncogenic transcripts (e.g., SRF) from miRNA-directed transcript downregulation [156]. This demonstrates that regulation of transcripts relies on more than just binding, and in some cases relates to which family member interacts with the RNA. Could these shifts in family member expression represent a shift in how RNA stability is managed in neuroblastoma cells? This could alter the location of transcripts and their susceptibility to miRNA attack. The probable involvement of LIN28B and DICER in neuroblastoma initiation suggests that the activity of miRNAs plays a substantial role in the development of neuroblastoma. Because many RBPs influence miRNA activity, translation and splicing, protein levels may differ from the mRNA levels observed. This highlights one limitation of this study, that is, we are using mRNA abundance for analyses. Many RBPs may be differentially expressed at the protein level or have isoforms with different roles. We noted that DHX9 has reported protein localisation changes which influence MNA neuroblastoma cell phenotype. One surprising aspect of the RBPs discussed in this paper is that some proteins are found in the serum of patients and thus are putative non-invasive biomarkers of neuroblastoma (e.g., MKRN3, DHX9) and, possibly, differentially secreted into serum. Auto-antibodies against IGF2BPs might also offer a diagnostic strategy which is yet to be explored in neuroblastoma. In histological analysis of tumours, the FBL protein pattern may indicate the use of the rRNA synthesis inhibitor CX-5461. Moreover, many RBPs could be used in prognostic signatures and as protein markers, thus having potential as biomarkers. 

By identifying genes that are important in neuroblastoma as RBPs, it may allow better mechanism-of-action studies in neuroblastoma. The literature discussing their roles, especially binding transcripts (which are useful for drug discovery) is scant, yet promising. This is surprising, as LIN28B, perhaps the best characterised RBP in neuroblastoma, can induce neuroblastoma in mice and potentially in some rare human cases as well. The remarkable effects of LIN28B and the other reported RBPs in neuroblastoma, as well as the great potential of our elucidated factors, endorse further study of RBPs in neuroblastoma. The rationale is even stronger when considering the available methods in high-throughput RNA-binding analysis (yet to be performed in neuroblastoma) and computer-aided drug discovery, which has a track-record of finding inhibitors of oncogenic RNA–protein interactions. With more than a third of RBPs having prognostic significance in neuroblastoma, and many having inhibitors and/or druggable structures, they undoubtedly present an opportunity for developing novel therapies against neuroblastoma. Moreover, due to their association with neurodegenerative diseases and viral replication, inhibitors of RBPs have clinical relevance outside cancer. Targeting RBPs is clearly a strategy one can engage with to target key RNAs that we know are crucial in neuroblastoma oncogenic signalling, and perhaps indirectly drug MYCN signalling. To use a phrase that is overused by researchers, but one that is nonetheless certainly apt here, the reported functions of RBPs in neuroblastoma represent just the “tip of the iceberg” of expected RBP roles in neuroblastoma. 

## 4. Materials and Methods

### 4.1. Differential Expression Analysis

The RBP gene list from Gerstberger et al. 2014 [17] was used to filter for RBPs and then compare *MYCN* amplification status of tumours, or stages 4 and 4S, alive and dead high-risk patients. Both are displayed as a volcano plot using the two groups plotter option of the R2 visualisation tool (https://r2.amc.nl) with the Kruskal rank-based test. The SEQC-498 custom dataset was used on the R2 platform (GEO accession: GSE49710) [20]. Outlying genes and genes of putative relevance in cancer and drug targeting were annotated where possible. mRNAs had to have a minimum of 50 expression units and be present in 100 tumours analysed for Figure 1A, and 50 tumours for Figure 1B and Figure 2. Five tumours were excluded from the *MYCN* amplification status comparison because this information was unknown, and FDR correction of p-values (FDR) was undertaken. Similarly, a second cohort was used in an attempt to confirm the results of the first cohort. This used the Versteeg—88 dataset, and the same gene list was applied for the figure as for that SEQC dataset. 

### 4.2. Kaplan–Meier Significance Scanning with the RBP Gene List

The RBP gene list from Gerstberger, et al. 2014 [17] was used to analyse the SEQC-498 custom dataset, with the Kaplan–Meier scan function. This was performed to find the RBP with the greatest significance of difference between the Kaplan–Meier curves (if significance could be reached), and a multiple testing correction was employed in the analysis. This indicated RBPs with prognostic value in the cohort. 

### 4.3. Heat Map of Expression and Dependency in Neuroblastoma Cell Lines

The expression of candidate RBPs was analysed with the R2 platform (https://r2.amc.nl) in neuroblastoma cell lines from the CCLE dataset (GEO accession: GSE36133, [149]) and in a large primary tumour cohort (SEQC Custom GEO accession: GSE49710). Stages 1, 2 and 3 were combined with the displayed average expression. Group sample sizes were as follows: 1–3 (*n* = 262), stage 4 (*n* = 183), stage 4S (*n* = 53), single *MYCN* (*n* = 401) or MNA (*n* = 92) tumours, respectively. The dependency scores were taken from the Broad Institute Cancer Dependency Map, CRISPR Avana dataset 17Q4 [150], and the average scores from all neuroblastoma cell lines (*n* = 11) are displayed next to the average dependency score over all other cancer entities (*n* = 330). Data can be downloaded from https://figshare.com/articles/Broad_Institute_Cancer_Dependency_Map_CRISPR_Avana_dataset_17Q4/5520160/1. The lower the score, the higher the selective effect of genetic knockout of the targeted genes on cell viability or growth in a cell population. GraphPad Prism 8.4 was used to create the heat maps. 

## Figures and Tables

**Figure 1 ijms-21-05098-f001:**
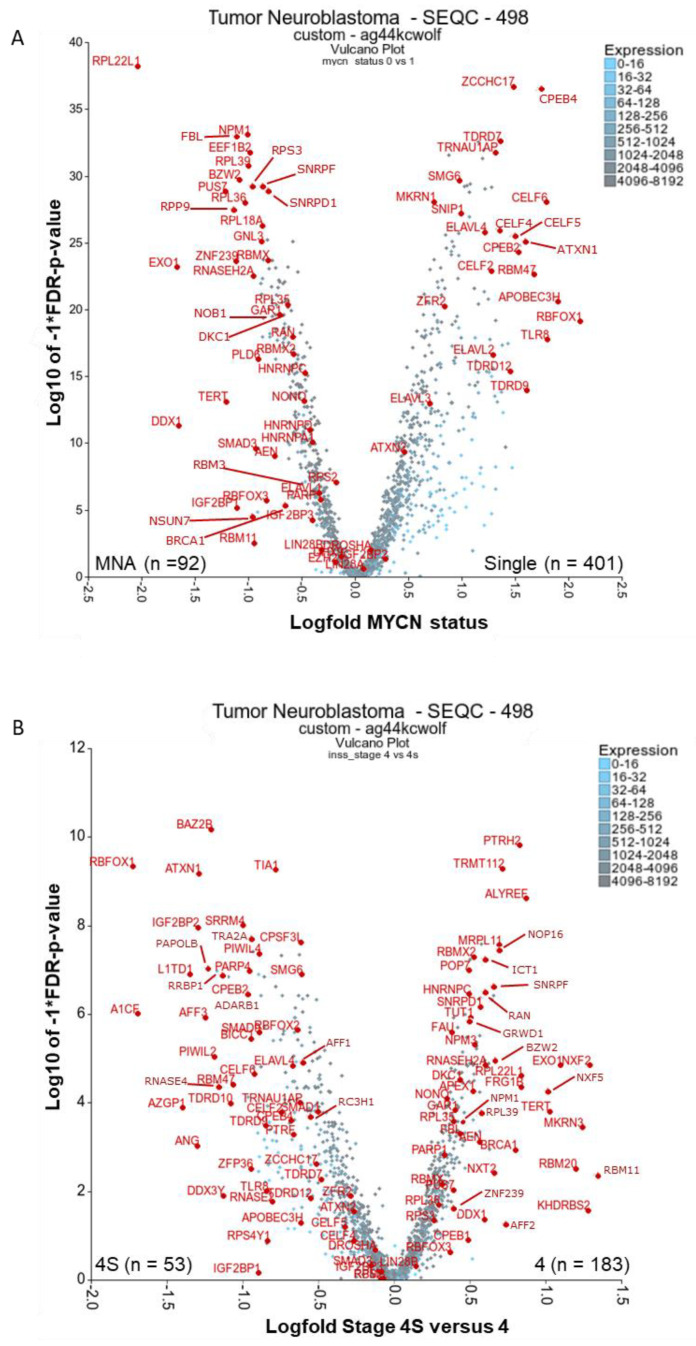
Differential expression of RNA-binding proteins (RBPs) in clinically relevant neuroblastoma groupings. The RBP gene list from Gerstberger, et al. 2014 [17] was used to filter for RBP mRNAs and then compare (**A**) *MYCN* status of tumours or (**B**) Stages 4 and 4S. Both the analyses were displayed as a volcano plot using the two groups plotter option of the visualisation tool R2 (https://r2.amc.nl). SEQC-498 cohort with the custom normalisation option was used on the R2 platform (GEO accession: GSE49710). Outlying genes and genes of putative relevance in cancer and drug targeting have been annotated where possible. mRNAs had to have a minimum of 50 expression units and be present in 100 tumours analysed for (**A**) and 50 tumours for (**B**). Five tumours were excluded from *MYCN* status analysis because their *MYCN* status was unknown. Statistical analysis was conducted with the Kruskal rank-based test, and false discovery rate (FDR) correction of p-values was performed. MNA = *MYCN* amplified, single = non-amplified *MYCN* status.

**Figure 2 ijms-21-05098-f002:**
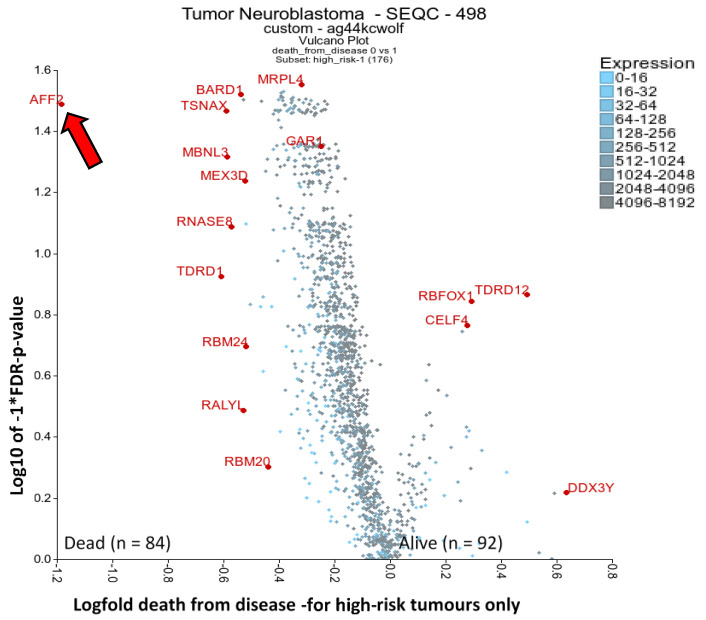
Differential expression of RBPs, analysing occurrence of death (from disease) in high-risk disease. The RBP gene list from Gerstberger et al. 2014 [17] was used filter for RBPs and then compare A) high-risk tumours only, comparing surviving or dead individuals. This was displayed as a volcano plot using the two groups plotter option of the R2 visualisation tool (https://r2.amc.nl). The SEQC-498 cohort with the custom normalisation option was used on the R2 platform (GEO accession: GSE49710). Clearly outlying genes have been annotated where possible. mRNAs had to have a minimum of 50 expression units and be present in 50 tumours analysed. Statistical analysis was conducted with the Kruskal rank-based test, and FDR correction of p-values was performed. The arrow highlights AFF2 next to *y*-axis.

**Figure 3 ijms-21-05098-f003:**
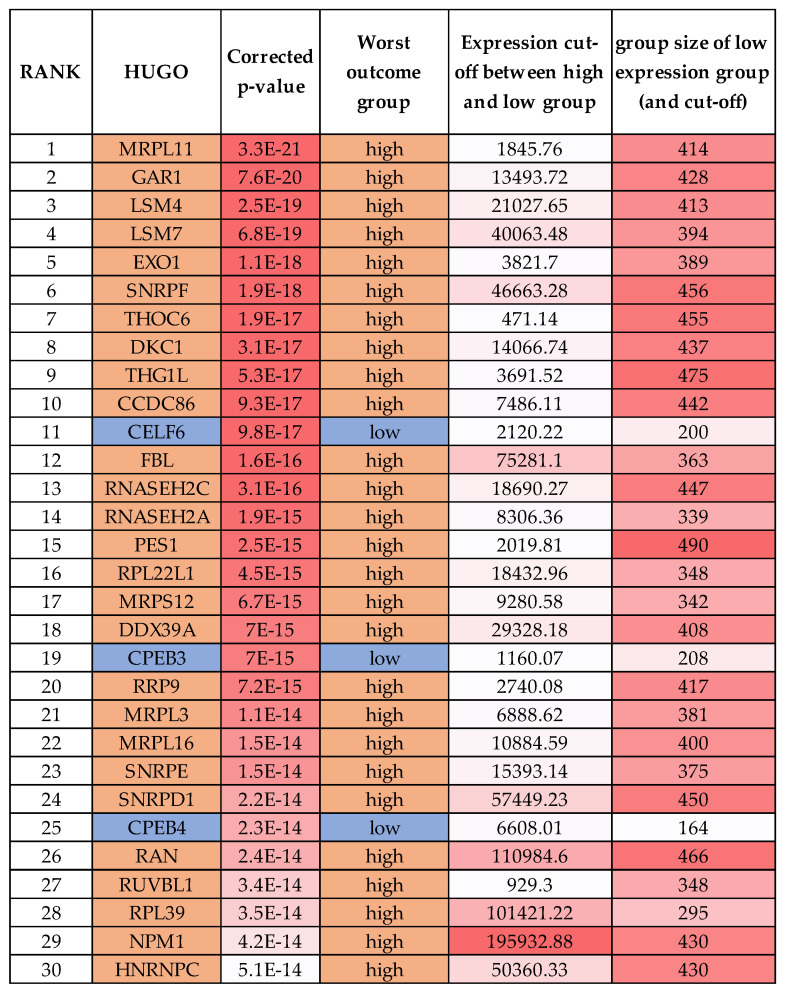
Top RBPs with the highest prognostic significance by the Kaplan scan method in the R2 visualisation tool.

**Figure 4 ijms-21-05098-f004:**
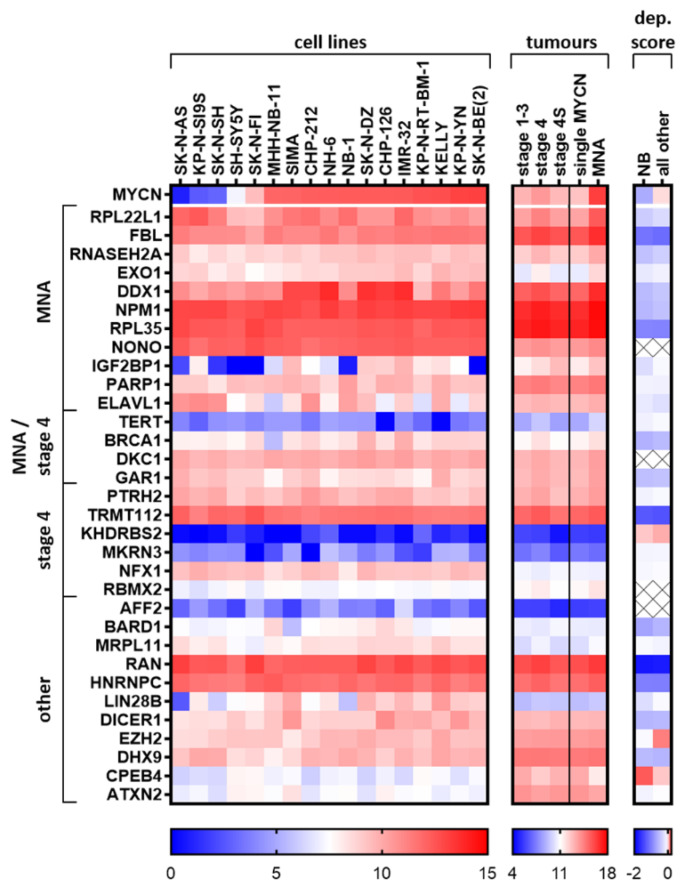
Expression of candidate oncogenic RBPs in neuroblastoma cell lines and in tumour groupings. Broad Institute Cancer Cell Line Encyclopedia (CCLE) neuroblastoma cell lines (GEO accession: GSE36133) and the SEQC primary tumour dataset (GEO accession: GSE49710) were analysed with the R2 platform for expression of candidate RBPs and GraphPad Prism was used to display the data. Cell lines were ordered by increasing *MYCN* expression. Dependency score (dep.score) averages were produced from the data provided by the Broad Institute Cancer Dependency Map, CRISPR Avana dataset 17Q4 [150]. The lower the score, the more essential the gene is for cell survival and/or proliferation. Positive scores indicate increased growth rate/cell survival compare to the parental cells when the gene is genetically deleted. Because sex chromosomes were excluded as part of the CERES (computational correction of copy-number effect in CRISPR-Cas9 essentiality screens) normalisation of the dependency score (part of the adjustment for genomic DNA copy number), CERES-excluded values are marked with an X (most of these excluded genes are located on the X chromosome). The last two genes displayed, *CPEB4* and *ATXN2*, are not considered oncogenic and we consider them tumour-suppressor-like RBPs, and these are included in the heatmaps to improve contrast. NB = Neuroblastoma cell lines.

**Figure 5 ijms-21-05098-f005:**
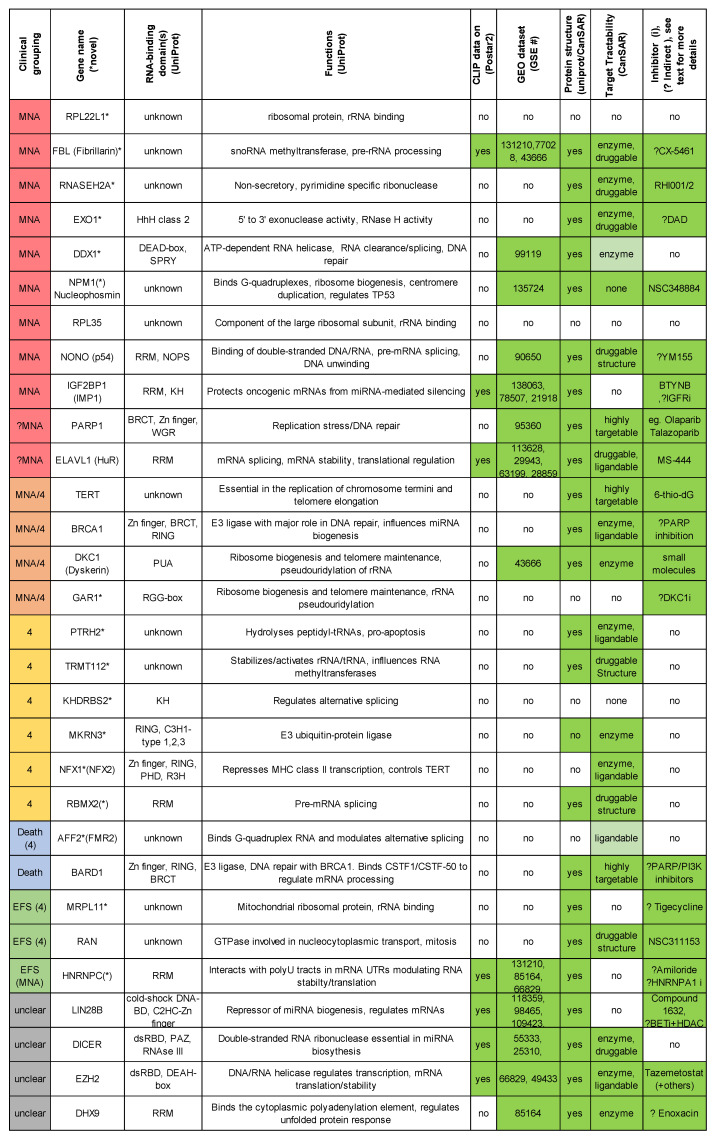
Selected novel and reported oncogene-like RBP characteristics with crystal structures and inhibitors listed. Green in columns on the right indicate favourable characteristics for drug development.

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
