# Peer review of "Identification of RNA-Binding Proteins as Targetable Putative Oncogenes in Neuroblastoma"

_ijms, 2020, doi:10.3390/ijms21145098_

Round 1

Reviewer 1 Report

The manuscript by Jessica L. Bell et al. focus on identification of RNA-binding proteins (RBP) using the RBP gene list from Gerstberger, et al. 2014 and then performed the survival analysis. The authors also performed Kaplan Meier Significance Scanning with the RBP gene list from Gerstberger, et al. 2014 and R2 platform for heat map of expression and dependency in neuroblastoma cell lines. This manuscript has no novelty since the authors only reanalyzed the RBP gene list from Gerstberger, et al. 2014 and no further experimental validation. There are no deep results provided in the manuscript. The authors only described the RBP according to the analyzed data and didn’t offer any new information for neuroblastoma community. Additionally, there are no any experiments to validate the analyzed results, so I don’t think they can conclude that “we confirmed the targeting of RBPs as an alternative, but nonetheless promising therapeutic strategy in neuroblastoma treatment”. As I mentioned above, I decide to reject this manuscript.

Author Response

  1. This manuscript has no novelty since the authors only reanalyzed the RBP gene list from Gerstberger, et al. 2014 and no further experimental validation.

Response: To the best of our knowledge this is the first time a review, conference abstract, book chapter, or article has tackled the topic of RBPs in neuroblastoma. If you have another example of such a paper, please inform us. The Gerstberger, et al. 2014 list was created very generally- it is not cancer-centric. It is just a list of RBPs in humans. This is the first time Gerstberger, et al. 2014 list has been applied using these methods, dataset and in the context of neuroblastoma. The work we have produced is an important resource and can be the starting point for future reviews or research papers covering the topic investigated.

We actually originally wanted to write a “review” on RBPs and neuroblastoma, but as most RBPs reported in neuroblastoma or not acknowledged as RBPs in the papers, proteins are not acknowledged as RBPs instead of searching 1483 different genes and their synonyms on PubMed we instead decided to use differential expression analysis using RBP consensus list to see what RBPs could be clinically important and also use these top genes to do literature searches, this allowed us to find exponentially more RBPs, then what we found already using "RNA-binding protein" and "neuroblastoma" as search terms on PubMed and google, or by searching for commonly cited RBPs and neuroblastoma. Because the information we found was highly novel and more interesting than expected, we wanted to include those analyses in the paper too, so others who work of neuroblastoma or those genes cited would be alerted to the paradigms we found. Yes we did not do wet-lab work. It is not a requirement of research articles to always include experimental work. I can find several papers on IJMS where existing lists or bioinformatics tools were used e.g. PMID: 32471155. We also saw in the special issue we have submitted to, PMID: 32050583, which had protein binding quantification with no functional validations offered. By submitting it as a research paper we have the structure to include to methods we used, and people can more easily understand how the work was done. Functional validation is not feasible with the 10 days we have to respond to the decision.

We can ask the editor if we can change the article type to review if this would be more acceptable to you.

I have changed the line you mentioned from the abstract to now read. Although not all RBPs appear suitable for drug design, or carry prognostic significance, we show that several RBPs have strong rationale for inhibition and mechanistic studies, representing an alternative, but nonetheless promising therapeutic strategy in neuroblastoma treatment.

  1. The authors only described the RBP according to the analysed data and did not offer any new information for neuroblastoma community.

Response: We have given the neuroblastoma community a list of oncogenes that should be studied in greater detail and a list showing the availability of drugs for these. It was not the purpose of the article to validate all the RBPs that were found but to acknowledge their reported roles and highlight those that should be investigated in future, and if any inhibitors exist. The neuroblastoma community has been extremely focused on transcriptional factors as therapeutic strategies, this article offers an alternative path that could also be tested.   

Reviewer 2 Report

The paper is overall interesting and trying to utilize a novel approach to identify new therapeutic targets.  It is a bit difficult to read as it feels like a very long list of possibilities.  There is an attempt to prove feasibility at targeting the various RBPs, but even that has many question marks or untargatable RBPs.  The paper would be much more impactful if it included a functional validation of the top targets as opposed to a long list of many possibilities.  

Author Response

We actually originally aimed to write a “review” on RBPs and neuroblastoma, but as most RBPs reported in neuroblastoma or not acknowledged as RBPs in the papers, proteins are not acknowledged as RBPs instead of searching 1483 different genes and their synonyms on PubMed we instead decided to use differential expression analysis using RBP consensus list to see what RBPs could be clinically important and also use these top genes to do literature searches, this allowed us to find exponentially more RBPs, then what we found already using "RNA-binding protein" and "neuroblastoma" as search terms on PubMed and google, or by searching for commonly cited RBPs and neuroblastoma. Because the information we found was highly novel and more interesting than expected, we wanted to include those analyses in the paper too, so others who work of neuroblastoma or those genes cited would be alerted to the paradigms we found. Yes we did not do wet-lab work. It is not a requirement of research articles to always include experimental work. I can find several papers on IJMS where existing lists or bioinformatics tools were used eg. PMID: 32471155. We also saw in the special issue we have submitted to, PMID: 32050583, which had protein binding quantification with no functional validations offered. The paper is consistent with other work published in this journal. By submitting it as a research paper we have the structure to include to methods we used and people can more easily understand how the work was done. But this explains why some subheading are a bit long. We were reviewing what is known on the RBPs elucidated or previously reported. I agree with you very much so, that it would be more impactful if we did functional validation, but this was not the purpose of the paper, we aimed that other researchers inspired by our work go on to do this. The purpose of the paper was to highlight an alternative regulatory level apart from transcription that could be potentially targeted by drugs. Functional validation is not feasible with the 10 days that we have to respond to the decision.

Please indicate if we should ask the editor if we can change the article type to review (with a methods section) if this would be more acceptable to you or if you could be content with the current structure.

Yes, not all those proteins identified are currently druggable, however enough proteins have existing or druggable structures that are yet to be investigated to provide a compelling paradigm. I was surprised by the amount of “top” proteins we could identify drug strategies for. With any list of genes in a family or grouping not all will be easily targetable, this does not rule out a group of genes as targetable options. However, I understand where you are coming from.

I have changed the abstract to reflect your concerns.

I have changed the line you mentioned from the abstract to now read.

Although not all RBPs appear suitable for drug design, or carry prognostic significance, we show that several RBPs have strong rationale for inhibition and mechanistic studies, representing an alternative, but nonetheless promising therapeutic strategy in neuroblastoma treatment.

Note also changes to lines 603-605.

Not all candidate oncogenic RBPs have structures which could support current drug design modelling, these may have to be targeted indirectly or by an alternative strategy.

Reviewer 3 Report

In the present manuscript entitled- Identification of RNA-binding proteins as targetable putative oncogenes in neuroblastoma-by, Jessica L. Bell and colleagues investigated the online neuroblastoma database on RNA binding proteins' expression and their association with MYCN amplification, DNA copy number, risk status, etc. This article provided a lot of information on the role of several RNA binding proteins in various pathways, including ribosome biogenesis, telomerase maintenance, MYCN association, microRNA biogenesis, and patient’s death.

Here are my comments.

(1) Why the authors chose to explore only one database as there are multiple sets of data available on the R2 database. Therefore, I encourage authors to explore author databases similar to the present findings and put them all together to get a clear-cut conclusion.

(2) Kaplan survival curves of the highlighted RNA binding proteins data also important.

(3) Tables and Figures are not visible.

(4) This manuscript requires moderate English edits.

Author Response

(1) Why the authors chose to explore only one database as there are multiple sets of data available on the R2 database. Therefore, I encourage authors to explore author databases similar to the present findings and put them all together to get a clear-cut conclusion.

Response: We used the SEQC dataset because it offered the largest number of samples with transcriptomic results with patient survival data. Unfortunately many of the other data sets of comparable size especially Kocak as far as I know (Pubmed link: 23579273_ ) and some others have a large number of samples in common and are not independent. I see colleagues and papers citing “independent cohorts” falsely often, and have been warned by co-authors that many cohorts on R2 tool are not totally independent cohorts. The sizes of the other cohorts apart from Kocak and SEQC are too small to do high-risk only analyses as performed in figure 2. I have written to R2 to get advice as to which cohorts could be appropriate. I had a look at Asgharzadeh – 249 cohort from the TARGET dataset, this gave similar results to what is displayed in the paper. I can repeat Figure 1 analysis, if you still need this it in the next 3 weeks.

(2) Kaplan survival curves of the highlighted RNA binding proteins data also important.

Response: Yes, I agree with you. Table one and supplement has Kaplan Meier information in a table form. This is the output style of the Kaplan scan analyses.

Please indicate if you need the actual Kaplan Meier graphs or if the tabular form in the paper is sufficient. The tabular form, allowed 30 RBPs to be represented on one page, it reflects the same information as shown in the graphs.

(3) Tables and Figures are not visible.

Response: Yes, I saw this too! Sorry. When IJMS did the formatting and merged the figures and text into the one document, quality was lost. I informed the editor within 8 hours of getting their formatted version and sent a better version, with better quality figures and weird underlining/bold- inserted errors removed. Unfortunately, this was not the version forwarded to reviewers. This is the first time, I have submitted a paper with this journal, and I know now that even though you can submit figures and text separately, its better to submit a merged file and control for image quality yourself.

(4) This manuscript requires moderate English edits.

Response: We have corrected those I could see with an additional read through such as DROSHER, and should be DROSHA. Hopefully, you find it improved.

Round 2

Reviewer 2 Report

While this paper brings up potential new therapeutic targets, there is no validation and only a single data set is utilized.  I don't feel that the reviewers' comments were adequately addressed with the revision.  This study likely needs some work to strengthen it before publication.

Author Response

We have added volcano plot analyses of a second cohort using similar methods as used for figure 1. This can be found in Figure S2. The majority of RBPs we have pointed out as significantly differentially expressed in Figure 1 show similar trends in the second cohort.

Reviewer 3 Report

The authors revised this manuscript per the reviewer’s comments and now looks much better.

Author Response

Thank you